# Determining the interaction status and evolutionary fate of duplicated homomeric proteins

**Saurav Mallik**[ID], **Dan S. Tawfik**[ID]*

Department of Biomolecular Sciences, The Weizmann Institute of Science, Rehovot, Israel

* dan.tawfik@weizmann.ac.il

## Abstract

Oligomeric proteins are central to life. Duplication and divergence of their genes is a key evolutionary driver, also because duplications can yield very different outcomes. Given a homomeric ancestor, duplication can yield two paralogs that form two distinct homomeric complexes, or a heteromeric complex comprising both paralogs. Alternatively, one paralog remains a homomer while the other acquires a new partner. However, so far, conflicting trends have been noted with respect to which fate dominates, primarily because different methods and criteria are being used to assign the interaction status of paralogs. Here, we systematically analyzed all *Saccharomyces cerevisiae* and *Escherichia coli* oligomeric complexes that include paralogous proteins. We found that the proportions of homo-hetero duplication fates strongly depend on a variety of factors, yet that nonetheless, rigorous filtering gives a consistent picture. In *E. coli* about 50%, of the paralogous pairs appear to have retained the ancestral homomeric interaction, whereas in *S. cerevisiae* only ~10% retained a homomeric state. This difference was also observed when unique complexes were counted instead of paralogous gene pairs. We further show that this difference is accounted for by multiple cases of heteromeric yeast complexes that share common ancestry with homomeric bacterial complexes. Our analysis settles contradicting trends and conflicting previous analyses, and provides a systematic and rigorous pipeline for delineating the fate of duplicated oligomers in any organism for which protein-protein interaction data are available.

## Author summary

About half of all proteins assemble as oligomers, either by self-interaction (homomers) or via interaction with another protein (heteromers). The latter can be unrelated, yet, quite commonly, the interacting proteins are paralogs, namely two genes that arose by gene duplication. Indeed, while a homomer is encoded by a single gene, heteromers demand two genes as a minimum. Duplication can therefore yield two discrete homomeric complexes or a single heteromer. Do paralogs tend to retain the ancestral homomeric interaction, or do they mostly diverge into heteromeric complexes? Despite several studies

**Data Availability Statement:** All the relevant data are available as Supplementary Data files (S1–S6 Data).

**Funding:** This work is supported by the Estate of Mark Scher, and by Israel Science Foundation

grant No. 2575/20. S.M. was supported by PBC-VATAT Postdoctoral Fellowship, provided by the Council for Higher Education, Israel. D.S.T. is the Nella and Leon Benoziyo Professor of Biochemistry at WIS. The funders had no role in study design, data collection and analysis, decision to publish, or preparation of the manuscript.

**Competing interests:** The authors have declared that no competing interests exist.

addressing this question, to date, we lack a systematic, rigorous approach for delineating the oligomeric fates of paralogs on an organism scale. To this end, we developed a new pipeline for analysis of molecular interaction databases that includes various filtering steps and unambiguous definitions of all possible oligomeric fates. Applying this method to *Escherichia coli* and *Saccharomyces cerevisiae* we noted that paralogous pairs tend to remain homomeric in the former while in the latter heteromeric complexes dominate. We consequently note a systematic trend of homomeric bacterial proteins diverging into heteromeric complexes in eukaryotes.

## Introduction

It is estimated that more than half of all proteins form oligomers. Oligomerization is thus ubiquitous and central to protein stability, function and regulation. Duplication is also ubiquitous and hence serves as the main source of new genes/proteins, as manifested by nearly half of all genes in a given genome being paralogs [1]. The duplication of genes encoding an oligomeric protein is of particular interest–the ancestral function may diverge alongside the oligomeric state thus providing new opportunities for evolutionary innovation [2–5].

Our analysis examined the divergence of homomers. By parsimony, the ancestors of both homomers and heteromers are homomers, as homomers are encoded by a single gene. Indeed, proteins have an inherent tendency to self-interact, and initially promiscuous self-interactions can be readily amplified by mutations to generate tightly bound homo-dimers and also larger homo-oligomers [6]. Upon duplication of a gene encoding a homomeric ancestor, and acquisition of the very first mutation(s), in either the original gene or its new copy, a statistical mixture of homo- and hetero-meric complexes would form [2] (**Fig 1A**, *i*). Over time, further evolutionary divergence may result in three possible scenarios: (**Fig 1A**, *ii*) loss of the capacity to cross-react and formation of two distinct homomeric complexes, or *obligatory homomers*; (**Fig 1A**, *iii*) loss of the homomeric interactions and formation of a heteromeric complex, or *obligatory heteromers*. Alternatively, the interaction pattern may diverge asymmetrically–while one paralogue is kept as homomer, the other gains a completely new interaction partner (**Fig 1A**, *iv*): *hetero-others*). Other scenarios may occur, *e.g.*, loss of homomeric interactions in both copies, or divergence into monomers that do not interact with any other protein; however, these scenarios are intractable on a genome-scale (by parsimony, their ancestors cannot be assumed to be a homomer) and are probably relatively rare.

Individual cases following all these four scenarios are known. What remains unclear, however, is which fate is the most likely? Does protein function, or the source organism, for example, affect which fate dominates? Genome-scale studies [2–4] attempted to address the relative frequencies of these scenarios in model organisms, but their conclusions are inconsistent. Analyzing human, *Arabidopsis*, yeast and *E. coli* protein-protein interaction (PPI) data, [3] reported that most oligomeric paralogs diverged to form obligatory homomers. However, analysis of yeast, worm and fly, using both PPI data and oligomers of known structure, [2] indicated that heteromeric interactions dominate, a conclusion recently supported by [4] who analyzed yeast PPI data. We compared these studies and observed that these inconsistencies relate to three major factors. First, different evolutionary scenarios were examined in different studies–*e.g.*, ref.[3] did not consider the mixed homo/heteromers, and essentially none of these studies [2–4] consider hetero-others. Second, different interaction datasets were analyzed ranging from X-ray crystallographic structures (*e.g.*, ref.[2]) to high-throughput PPI data (*e.g.*,

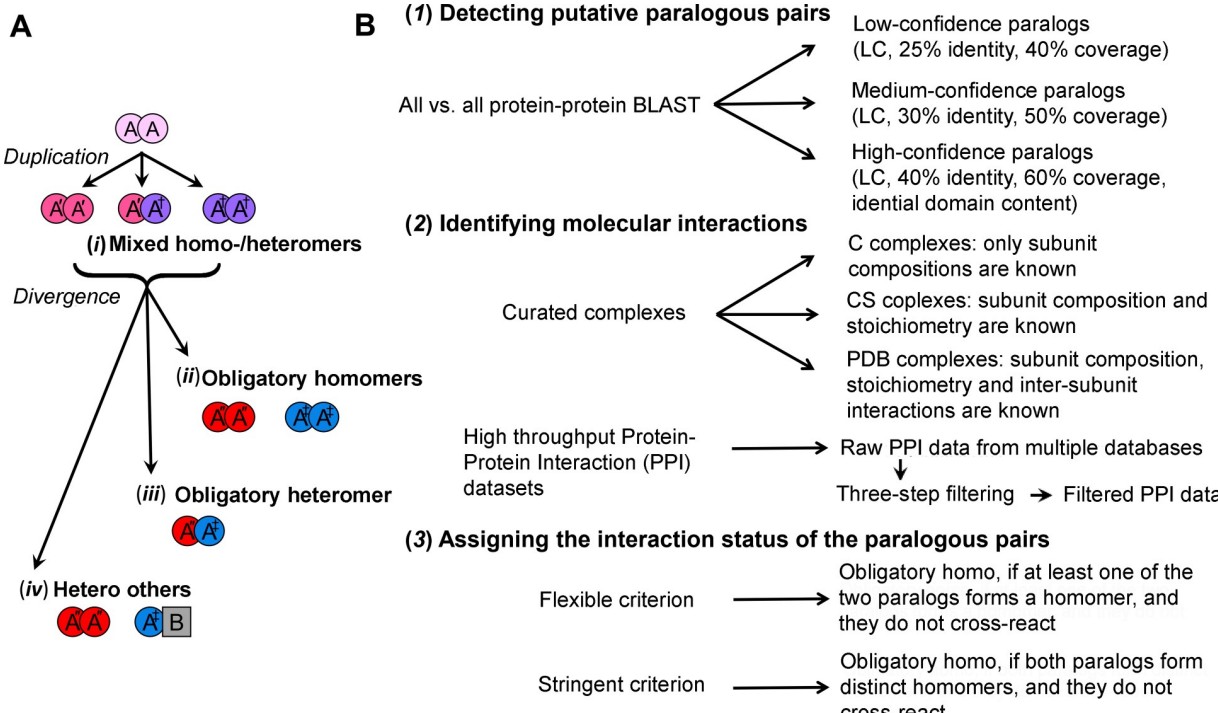

**Fig 1. The potential evolutionary fates of duplicated homomeric proteins and the analysis pipeline for identifying them. (A)** Duplication of a gene encoding a homomeric protein, and the emergence of the first mutation(s), leads to a statistical mixture of homo- and heteromeric complexes (*i*). Upon further divergence, three outcomes may arise: two distinct homomeric complexes (*ii*), a heteromeric complex involving both paralogs (*iii*), or loss of homomeric interaction in one copy, and gain of new interacting partners in the other paralog (*iv*). **(B)** Our analysis aimed to identify these four different evolutionary fates. It comprised three steps: (*1*) The genomes of *E. coli* and *S. cerevisiae* were each scanned to identify all possible paralogous protein pairs. These pairs were classified into three categories with increasing confidence of paralog assignment (note that all categories in our analysis are inclusive, *i.e.*, low-confidence paralogs include the medium-confidence ones, and the medium include the low-confidence pairs). (*2*) Interactions of these paralogs were identified and classified to homo- and heteromeric ones. Macromolecular complexes were collected from the Protein Data Bank (PDB complexes, inter-subunit interactions were obtained from crystal structure data) and the Complex Portal database (CS and C complexes, inter-subunit interactions were predicted from the PPI data). The *S. cerevisiae* PPI data were extracted from seven databases, and the *E. coli* data from eight databases. The raw PPI data were filtered using various criteria to exclude potential false-positives. (*3*) Finally, based on the identified interactions, the paralogous pairs were assigned to one of the four potential fates (*i-iv*, panel A) with either a *flexible* or a *stringent criterion*.

refs.[3,4]). Third, the divergence modes were assigned in an incongruous fashion. Following the definition of obligatory homomers, ref.[2] demanded both paralogs to be assigned as homomers; but others, refs.[3,4], for example, sufficed with identifying just one paralog as homomer.

Taking advantage of the extensive characterization of *Saccharomyces cerevisiae* and *Escherichia coli* macromolecular complexes, we investigated the potential evolutionary fates of their duplicated homomeric proteins. We systematically varied the stringencies of assigning paralogous pairs, of filtering molecular interaction datasets, and of assigning the divergence modes, and examined how these parameters affect the assigned proportions of homo-hetero divergence events. In *S. cerevisiae*, when stringent criteria were applied, a consistent picture arose, indicating that contrary to a previous analysis [3], 90% of duplications resulted in heteromeric complexes. In *E. coli*, however, it appears that paralogs are 5 times more likely to retain their ancestral homomeric interactions. We reconciled this difference by tracking down individual complexes and showing that complexes that are homomeric in *E. coli* have, upon duplication, diverged to heteromeric complexes in *S. cerevisiae*.

## Results

### A systematic approach to delineate the evolutionary fates of duplicated homomers

We analyzed the relative abundances of the four potential fates by examining the proteomes of *S. cerevisiae* and *E. coli* for which extensive interaction data exist. As the inconsistencies between previous works depict, this analysis presents biases at each one of its three steps (**Fig 1B**). In the 1$^{st}$ step, considering only paralogous pairs with high sequence coverage and identity would enrich closely related pairs that are more likely to be detected as mixed homo-heteromers. Conversely, assigning paralogous pairs with low coverage and identity might include cases where the changes in the divergence modes are due to loss or gain of entire domains rather than divergence of preexisting interfaces. To address this bias, in the 1$^{st}$ step, we classified the putative paralogous pairs into three groups going from low to high confidence of paralogue assignment (**Fig 1B, 1**).

In the 2$^{nd}$ step, structures of macromolecular complexes allow to assign interactions with high accuracy, but crystal structures in particular create a bias in favor of homomeric interactions [7]. High-throughput protein-protein interaction (PPI) data cover a much larger set of proteins, yet they can be noisy, and how these data are filtered would substantially influence the results. Beyond random noise, there are biases–for example, certain PPI methods cannot detect homomeric interactions (*e.g.* pulldown and MS identification of binding partners). We thus analyzed separately and compared the results from curated complex datasets (hereafter, *curated complexes*) and high-throughput PPI data (**Fig 1B, 2**). For the latter, PPI data were pulled together from different databases (**S1 and S2 Data**) and taken through three different filters to minimize false-positives. These databases encompass all reported interactions, including high resolution data, yet the high throughput data dominate, especially after the applied filtering.

Finally, in the 3$^{rd}$ step, the criteria for assigning the fates of paralogous pairs also matter. In principle, obligatory-homomers means that *both* paralogs were individually observed as homomers and that a cross-interaction was not observed (*stringent criterion*). Suffing with one paralog that forms a homomer would inevitably result in obligatory-homomers being the most frequent fate [4]. Further, as shown below, applying this *flexible criterion* results in assigning paralogs that actually diverged to hetero-others as obligatory-homo (**Fig 1A**, *iv* and *ii*, respectively). Thus, the divergence modes of the paralogous pairs were assigned applying both *stringent* and *flexible criteria* (**Fig 1B**, *3*). We subsequently examined the relative frequency of the four divergence modes, or fates, as a function of the stringency of analysis in each of the 3 steps.

Few clarifying notes regarding our analysis. We addressed paralogous pairs, *i.e.*, pairs of two genes that diverged from a common ancestor. In many cases, multiple paralogs exist that arose from two or more sequential duplications. Initially, we detected all potential pairs (**Fig 1B**, step-1). Then, by assigning the divergence modes, we defined the relevant paralogous pairs (with few exceptions in the mixed category (**Fig 1A**, *i*) where one protein can be part of more than one pair). Thus, unless otherwise stated, the statistics and below discussion relate to gene pairs. Additionally, given that some complexes comprise multiple pairs, statistics are also provided per complexes. Finally, our parsimonious assumption is that the pre-duplicated ancestor can be considered a homomer if at least one descendent paralog is a homomer, and also if both paralogs are present as a heteromer (as in [2,4]). The latter was subsequently confirmed by our analysis ('Yeast heteromeric paralogs diverged from bacterial homomeric ancestors').

The results of our analysis were distilled to **Fig 2** that presents the relative frequency of the four divergence modes given the dataset and stringency of analysis. The tables are arranged

**A**

| Assigning paralogs | Curated macromolecular complex data | | | Protein-Protein interaction data | | | | | | | | | | | |
|---|---|---|---|---|---|---|---|---|---|---|---|---|---|---|---|
| | | | | Raw data | | | Filter-1 | | | Filter-2 | | | Filter-3 | | |
| | | | | | | | Transposon Elements removed | | | Transposon Elements removed Reported in ≥ 2 databases Reported as both bait and prey | | | Transposon Elements removed Reported in ≥ 2 databases Reported as both bait and prey Filtered for Localization | | |
| | LC | MC | HC | LC | MC | HC | LC | MC | HC | LC | MC | HC | LC | MC | HC |
| **(i) Flexible critaria** | n = 707 | n = 428 | n = 170 | n = 3531 | n = 2761 | n = 2162 | n = 2080 | n = 1323 | n = 724 | n = 2013 | n = 1275 | n = 691 | n = 1449 | n = 862 | n = 408 |
| Obligatory homo | 57.4 | 58.6 | 65.3 | 50.1 | 54.1 | 61.4 | 30.3 | 27.4 | 26.9 | 28.2 | 24.3 | 21.4 | 31.7 | 27.6 | 23.8 |
| Obligatory hetero | 22.3 | 22.0 | 18.2 | 12.5 | 11.2 | 8.9 | 21.2 | 23.4 | 26.4 | 21.1 | 23.4 | 26.6 | 22.3 | 25.8 | 31.4 |
| Mixed homo/hetero | 2.8 | 3.0 | 3.5 | 19.8 | 18.1 | 13.3 | 32.5 | 36.0 | 36.5 | 32.9 | 36.6 | 37.5 | 25.5 | 29.4 | 30.4 |
| Hetero others | 17.4 | 16.4 | 12.9 | 17.6 | 16.6 | 16.5 | 16.1 | 13.2 | 10.2 | 17.8 | 15.7 | 14.5 | 20.5 | 17.3 | 14.5 |
| **(ii) Stringent criteria** | n = 337 | n = 198 | n = 67 | n = 2907 | n = 2303 | n = 1806 | n = 1745 | n = 1148 | n = 650 | n = 1655 | n = 1075 | n = 591 | n = 1152 | n = 713 | n = 349 |
| Obligatory homo | 10.7 | 10.6 | **11.9** | 39.4 | 45.0 | 53.7 | 16.9 | 16.4 | 18.6 | 12.6 | 10.2 | 8.1 | 14.1 | 12.5 | **10.9** |
| Obligatory hetero | 46.9 | 47.5 | **46.3** | 15.2 | 13.4 | 10.6 | 25.2 | 26.9 | 29.4 | 25.7 | 27.7 | 31.1 | 28.0 | 31.1 | **36.7** |
| Mixed homo/hetero | 5.9 | 6.6 | **9.0** | 24.0 | 21.7 | 15.9 | 38.7 | 41.5 | 40.6 | 40.1 | 43.4 | 43.8 | 32.0 | 35.5 | **35.5** |
| Hetero others | 36.5 | 35.4 | **32.8** | 21.4 | 19.8 | 19.7 | 19.2 | 15.2 | 11.4 | 21.6 | 18.6 | 16.9 | 25.8 | 20.9 | **16.9** |

**B**

| Assigning paralogs | Curated macromolecular complex data | | | Protein-Protein interaction data | | | | | |
|---|---|---|---|---|---|---|---|---|---|
| | | | | Raw data | | | Filter-1 | | |
| | | | | | | | Reported in ≥ 2 databases Reported as both bait and prey | | |
| | LC | MC | HC | LC | MC | HC | LC | MC | HC |
| **(i) Flexible critaria** | n = 215 | n = 138 | n = 89 | n = 1199 | n = 1191 | n = 363 | n = 868 | n = 387 | n = 73 |
| **Obligatory homo** | 86.5 | 87.0 | 87.0 | 46.6 | 46.5 | 28.1 | 66.1 | 62.8 | 37.0 |
| **Obligatory hetero** | 1.1 | 2.2 | 3.7 | 10.8 | 10.9 | 22.6 | 5.2 | 7.8 | 24.7 |
| **Mixed homo/hetero** | 3.4 | 2.2 | 1.9 | 18.2 | 18.3 | 36.4 | 5.0 | 8.5 | 26.0 |
| **Hetero others** | 9.0 | 8.7 | 7.4 | 24.4 | 24.3 | 12.9 | 23.7 | 20.9 | 12.3 |
| **(ii) Stringent criteria** | n = 57 | n = 43 | n = 31 | n = 907 | n = 902 | n = 319 | n = 662 | n = 306 | n = 64 |
| **Obligatory homo** | 50.9 | **58.1** | 61.3 | 30.8 | 30.7 | 17.6 | 55.6 | **52.9** | 28.1 |
| **Obligatory hetero** | 14.0 | **7.0** | 3.2 | 29.4 | 29.5 | 45.2 | 6.8 | **9.8** | 28.1 |
| **Mixed homo/hetero** | 7.0 | **7.0** | 9.7 | 10.7 | 10.8 | 24.4 | 6.5 | **10.8** | 29.7 |
| **Hetero others** | 28.1 | **27.9** | 25.8 | 29.0 | 29.1 | 12.8 | 31.1 | **26.5** | 14.1 |

**Fig 2. The distribution of divergence modes of *S. cerevisiae* and *E. coli* paralogous pairs.** The four divergence modes, obligatory-homo, obligatory-hetero, mixed and hetero-others, are described in **Fig 1A**. (**A**) The distribution of *S. cerevisiae* paralogous pairs in PPI data (*right panel*) and in curated complexes (*left panel*). Presented are the distributions for different stringencies of analysis, along its 3 steps (**Fig 1B**). Step-1, paralog assignment, is presented in *columns*, shaded in green, from low-confidence in pale green to high-confidence paralogs in dark green. Step-2, identifying interactions, also in *columns*, from white (raw PPI data) to dark grey (filter-3). Step-3, the divergence mode, is presented in *rows*–the top set of rows represent the *flexible criterion* (shaded in yellow), and the bottom rows the *stringent criterion* (dark yellow). The dominant divergence modes, or fates, are highlighted in darker shades of red. (**B**) The distribution of *E. coli* paralogous.

such that the darker the color, the higher is the stringency. The results given different stringencies of paralog assignment (Step-1) are presented in columns, going from low-confidence in pale green to high-confidence paralogs in dark green. Step-2, also in columns, from white (raw PPI data) to dark grey (Filter-3). Step-3, the stringency of assigning divergence modes, is presented in rows, with the top set of rows in yellow showing the *flexible criterion*, and the bottom, dark yellow rows indicating the *stringent criterion*. Finally, the dominant divergence modes, or fates, are highlighted in darker shades of red.

## Heteromeric interactions dominate yeast paralogs

For yeast, under stringent filtering, the results from curated complexes and from PPI largely converge, indicating that ~90% of yeast duplicates diverged to various heteromeric states. Specifically, stringent filtering of the PPI interactions (**Fig 2A**, Filter-3, dark grey columns), and applying the *stringent criterion* for assigning the divergence modes (**Fig 2A**, dark yellow rows),

indicated that only about one-tenth of the paralogous pairs diverged to obligatory-homomers. Given the consistency between the two datasets, and the noise origins and biases indicated by our analysis (elaborated below), we surmise that obligatory-homo are indeed a minority in yeast (~10%) and hetero-dominance is the reality (**Fig 2A**, numbers in bold, **S3 Data**). Within the three different hetero fates, the dominant fate is obligatory-hetero (about half of the pairs in the curated complexes, and a third in the PPI data where, as expected, a larger fraction of pairs was annotated as mixed).

If we were to count unique complexes instead of gene pairs, would the picture be different? Certain heteromeric complexes are composed of multiple paralogous proteins and these could shift the balance in favor of obligatory homomers (mostly ring-like complexes such as the proteasome; further addressed below). Nonetheless, analysis of complexes showed that, under the stringent filtering criteria, and for high-confidence paralogs, complexes comprising heteromers were nearly three-times more frequent than homomeric complexes (**Fig 3A**). Overall, we conclude that heteromeric interactions dominate yeast paralogs, regardless of whether we count paralogous pairs or unique complexes.

## Data biases and their mitigation

Our analysis also reveals various sources of error and bias, and how these could be mitigated. As expected, consistency of the two interaction datasets, curated complexes and PPI, fades away at lower stringency. Foremost, the 3$^{rd}$ step of the analysis, assigning the divergence modes, had a massive impact on the relative abundances of homo-hetero pairs. Assigning obligatory homomers using the *flexible criterion* (suffice that one paralog is a homomer and no cross-reaction) resulted in ~5-fold proliferation of obligatory-homomers in the curated complexes, and ~3-fold proliferation in the PPI data (**Fig 2A**, light yellow rows). The reason being that under the *flexible criterion*, hetero-others were assigned as obligatory-homo. Thus, cases that are quite abundant in yeast where one paralog kept the ancestral homomeric interaction and the other diverged to bind a completely new partner were not only ignored, but also misassigned.

Our analysis also reflects the homo- or hetero-biases that are inherent to the source of interaction data. The homo-dominance in the curated complexes dataset primarily stems from the known bias of crystal structures to detect homomers [7]; the hetero-dominance in the PPI dataset stems from certain high-throughput methods failing to detect homomeric interactions. Indeed, for a given a stringency with respect to the first two steps of the analysis (assigning paralogs, identifying interactions), homomers are more frequent in the curated complexes while heteromers dominate the PPI data (**Fig 2A**). However, these biases seem to be alleviated under the *stringent criterion*, as both the PPI and the curated complexes give a similar distribution of fates. Thus, consideration of all four evolutionary fates, namely including both mixed homo-hetero and hetero-others, is critical, as are adequate criteria to assign them (re the *stringent criteria*).

Two other elements seem to be critical for obtaining consistent results, both relating to the PPI data. Upon manual inspection we removed five long terminal repeat retrotransposon families, comprising a total of 90 proteins. These paralogous mobile genetic elements of viral origin [8,9] caused an inflation in the fraction of obligatory-homomers (~50%, that dropped to ~15% once removed). Further, once these retrotransposon proteins were removed (**Fig 2A**, filter-1), the homo-hetero fates in the PPI data converged with those in the curated complexes (**Fig 2A**, stringent criterion). Filtering of potential false-positives in the PPI data had a lesser effect. First, we applied a demand that interactions are reported in two different databases, and that interactions were detected with the protein pairs applied as both bait and prey (**Fig 2A**,

filter-2). The second source of false-positives are *in vitro* PPI interactions that do not occur *in vivo*. Obvious cases include interactions between proteins localized in different compartments (**Fig 2A**, filter-3). However, compared to the removal of retrotransposons these two filters had a minor effect.

Overall, we conclude that heteromeric interactions between paralogous pairs is the dominant fate in yeast, regardless of whether we count paralogous pairs or unique complexes.

### Homomeric interactions dominate *E. coli* paralogs

A similar analysis of *E. coli* indicated that in oppose to *S. cerevisiae*, for high-confidence paralogs, about 60% of the descendent pairs are obligatory-homers in the curated complexes compared to only 30% in the PPI data (**Fig 2B**, filter-1, HC, *stringent criterion*, **S4 Data**). However, this inconsistency is because the *E. coli* sample sizes for high-confidence paralogs are too small (**S1A Fig**). In yeast, filtering led to considerable reduction in sample sizes, yet these remained high even for high-confidence paralogs (**Fig 2A**). Further, the distribution is similar for high and medium-confidence, and with few exceptions even to the low-confidence (highest sample size, **S1B Fig**). This is not the case for the *E. coli* analysis. When more distantly related paralogs were removed (MC and HC columns), sample sizes decreased by >10-fold, compared to >3-fold in yeast. Indeed, in yeast, owing to the relatively recent whole genome duplication, high-confidence paralogs comprise ~60% of all detectable paralogs (1806/2907, **S1B and S1C Fig**), while in *E. coli* they comprise only ~35% (**S1B Fig**). Thus, it seems that medium-confidence paralogs better report the actual reality in *E. coli*.

Overall, considering the stringent criterion for assigning the divergence fates, the filtered PPI data and the curated complexes gave a consistent picture by which ~55% of the pairs comprise obligatory-homomers, for both medium- and low-confidence paralogs (**Fig 2B**, MC and LC). Further, as in yeast, homomers also dominated when complexes were counted (**Fig 3B**).

**A**

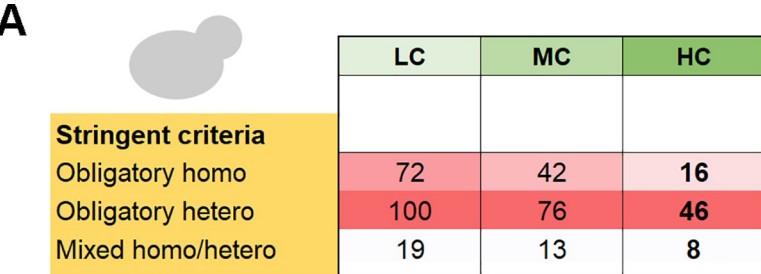

|  | LC | MC | HC |
|---|---|---|---|
| **Stringent criteria** |  |  |  |
| Obligatory homo | 72 | 42 | **16** |
| Obligatory hetero | 100 | 76 | **46** |
| Mixed homo/hetero | 19 | 13 | **8** |

**B**

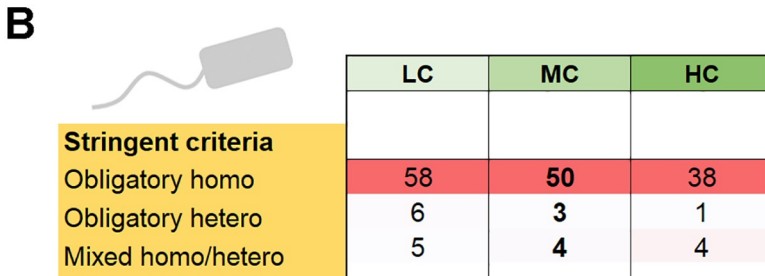

|  | LC | MC | HC |
|---|---|---|---|
| **Stringent criteria** |  |  |  |
| Obligatory homo | 58 | **50** | 38 |
| Obligatory hetero | 6 | **3** | 1 |
| Mixed homo/hetero | 5 | **4** | 4 |

**Fig 3. The distribution of complexes comprising homo- and heteromeric paralogs in *S. cerevisiae* and in *E. coli*.** This analysis was based on the curated complexes databases. The column annotations and color shades are the same as in **Fig 2**. (**A**) The numbers of unique *S. cerevisiae* complexes comprising paralogs assigned to the different homo/hetero divergence modes. Note that the different confidence levels for paralog assignment (LC, MC, HC) show that same trend as in **Fig 2B**, curated complex panel. (**B**) The same for *E. coli*.

Overall, it appears that retaining the ancestral homomeric interaction is the most likely fate of *E. coli* gene duplications.

Note that tuning the stringencies in the *E. coli* analysis had similar effects as in *S. cerevisiae*. Filtering the PPI for interactions reported in at least two databases, and as both bait and prey resulted in a higher fraction of obligatory-homomers. On the other hand, assigning the divergence modes with a *flexible criterion* resulted in overestimation of obligatory-homomers (and a corresponding drop in obligatory-heteromers).

### Yeast heteromeric paralogs diverged from bacterial homomeric ancestors

We observed the dominance of obligatory-homomers in *E. coli* (~50%) while in *S. cerevisiae* they comprise only ~10% of the duplicated oligomeric proteins, and in turn obligatory-heteromers comprise the majority. However, these two model organisms share common ancestry, as reflected in about one-third of *S. cerevisiae* proteins, many of which are mitochondrial proteins, harboring sequence signatures of bacterial origin [10]. We thus searched for the *E. coli* orthologs of the *S. cerevisiae* heteromeric paralogs, asking which are homomeric.

A systematic reciprocal BLAST was performed between all known *E. coli* homomers (n = 1033) and all *S. cerevisiae* obligatory-hetero and mixed paralogous pairs (n = 692; out of a total of 1152 LC pairs in the *stringent* categories, PPI dataset; **Fig 2A**). Following manual curation (see Methods), we identified about a third of the heteromeric yeast paralogous pairs that have *E. coli* homomeric orthologs (n = 235; **S5 Data**). Of these, nearly two-thirds, 153 pairs, relate to *E. coli* homomers that are singletons (*i.e.*, non-duplicated genes; a total of 52 proteins). By parsimony, these reflect cases of duplication and divergence of an ancestral bacterial homomer into paralogous heteromers in yeast. Remarkably, 42/52 of these *E. coli* proteins are metabolic enzymes that duplicated and diverged into heteromeric *S. cerevisiae* enzymes. In many such cases only one copy retained the catalytic activity whereas the other one evolved into a regulatory subunit. Examples include mitochondrial $NAD^+$-dependent isocitrate dehydrogenase complex [11], Trehalose Synthase Complex [12], the 20S proteasome core particle subunits [13], or the ATP-dependent 6-phospho-fructokinase complex [14,15]. Other enzymes, such as chaperonins, HSP70 chaperones, and DNA and RNA helicases appear to have gone through multiple duplications and contribute to the hetero-dominance in *S. cerevisiae*.

The remaining third, 82 yeast heteromeric paralogous pairs, are orthologous to 144 obligatory homomeric pairs in *E. coli* (**S5 Data**). These also relate to divergence of homomers to heteromers. What is unclear though is which of these genes duplicated independently in these two clades, and which one diverged to heteromers in an earlier bacterial ancestor. What is clear though is that the dominance of heteromeric paralogs in yeast is the result of homomers duplicating and preferentially diverging into heteromers.

## Discussion

With the obvious caveat of being based on two model organisms for which extensive protein interactions data are available, our analysis indeed suggests a continuous evolutionary process of bacterial homomeric proteins gradually duplicating and diverging into heteromeric proteins in eukaryotes. This ongoing evolutionary transition also validates our assignment of the fundamentally different divergence modes of paralogous pairs in *E. coli* and *S. cerevisiae* (**Fig 2**). Assuming *E. coli* and *S. cerevisiae* are representatives of bacteria and single-cell eukaryotes, the gene duplications that occurred in the eukaryotic lineage that diverged from bacteria via endosymbiosis [16,17] led to 5-fold decrease in the abundance of homomers among paralogous proteins. Further, because paralogous proteins comprise nearly half of the proteomes, this

phenomenon has led to a complete shift from the prevalence of homomers in prokaryotes to heteromers in eukaryotes [18,19].

The transition of homomeric prokaryotic complexes into eukaryotic heteromeric ones was previously noted for individual protein families, and especially for ring-like complexes such as DNA/RNA helicases [20,21], TCP complex subunits [22], proteasome [23,24] and exosome [25]. However, examining our dataset revealed that both ring-like and non-ring-like prokaryotic homomers evolved into heteromeric complexes in eukaryotes, and by a single or multiple gene duplications (**Fig 4**, **S5 Data**). Thus, the dominance of heteromeric paralogs in *S. cerevisiae* is not only because the ancestral homomers duplicated and diverged into heteromers, but also because heteromeric paralogs further duplicated and their descendants retained the heteromeric state.

For non-ring-like complexes, a single gene duplication typically results in a single eukaryotic heteromeric complex that may or may not retain the ancestral oligomeric order (total number of complex subunits). For example, *E. coli* DNA mismatch repair endonuclease MutL is a homodimer, and the yeast orthologue is a heterodimer [26] (**Fig 4**, *i*). On the other hand, the bacterial homo-dimeric isocitrate dehydrogenase [11] duplicated and diverged into a hetero-octameric mitochondrial isocitrate dehydrogenase in yeast [26]–namely, the oligomeric order changed from 2 to 8 (**Fig 4**, *ii*). In this case, duplication and divergence into a heteromer tendered the opportunity of evolving a new regulatory mode by diversifying one subunit, while the other subunit kept the catalytic activity.

As a prokaryotic non-ring-like homomer evolves into a heteromer in eukaryotes, multiple rounds of duplication may occur and the descendent paralogs retain the newly evolved

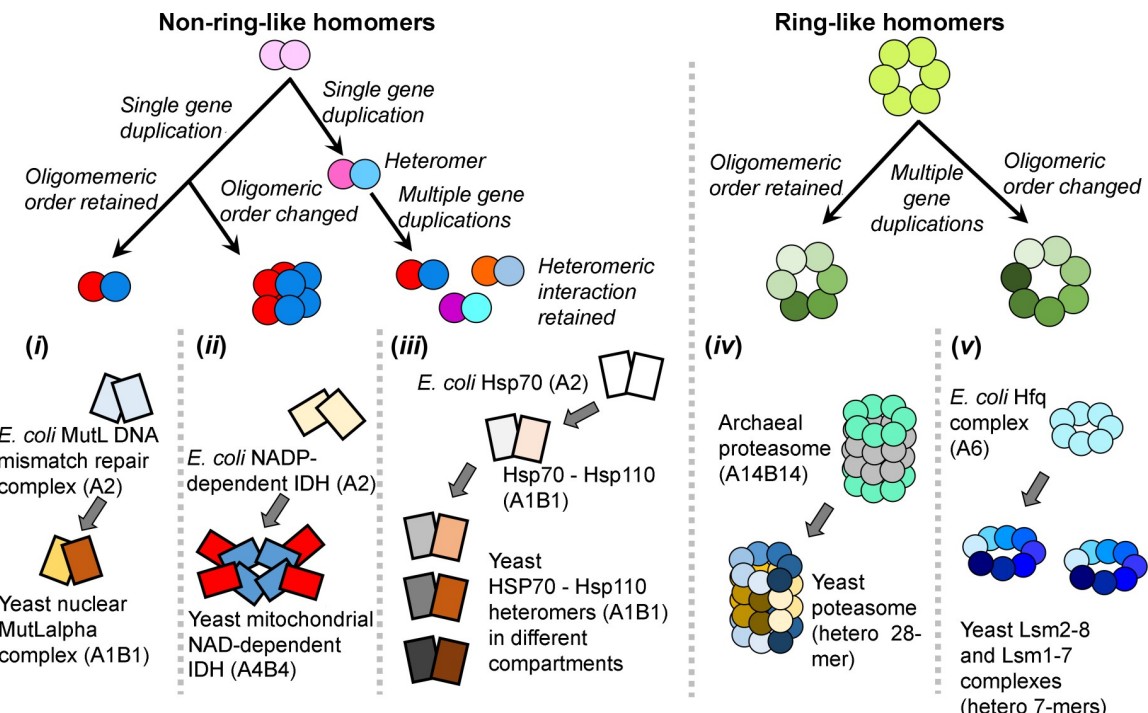

**Fig 4. Different modes of prokaryotic homomer to eukaryotic heteromer transition.** Gene duplication of an ancestral non-ring-like homomer may produce a heteromeric complex that may (*i*) or may not (*ii*) retain the ancestral oligomeric order (*i.e.*, the total number of subunits in the complex). After the first gene duplication and the subsequent emergence of a heteromeric interaction, multiple rounds of duplication may follow in which the descendant paralogs retain the heteromeric interaction (*iii*). For ring-like complexes, multiple rounds of intra-ring gene duplications result in heteromeric rings, while keeping (*iv*) or changing the ancestral oligomeric order (*v*). For each mode of transition, an example case is provided.

heteromeric interaction (**Fig 4**, ***iii***). For example, the bacterial homomeric Hsp70 that dupli-cated and diverged into Hsp110 co-chaperones in eukaryotes [27,28], and the *S. cerevisiae* genome encodes multiple copies of Hsp70 and Hsp110 that form distinct heteromers in differ-ent subcellular compartments [29].

For ring-like prokaryotic homomeric complexes (*e.g.*, helicase, protease, RNase and chaper-onins), homo-to-hetero transition predominantly also occurred while retaining the ancestral oligomeric order or modifying it (**Fig 4**, ***iv–v***). Complexes that have retained their ancestral oligomeric order (**Fig 4**, ***iv***) include the archaeal homo-hexameric MCM complex that became hetero-hexameric in eukaryotes [21], and the core proteasome alpha- and beta-rings that remained heptameric [23,24]. In contrast, the bacterial helicase homo-hexameric Hfq ring-complex [30] diverged to the hetero-heptameric Lsm1-7 and Lsm2-8 complexes in yeast (**Fig 4**, ***v***).

The above-described phenomena that underline homo-to-hetero transitions present some interesting questions. This transition needs to overcome the inherent self-interacting tendency of proteins, and eventually lead to incompatibility of the homomeric interactions. It is there-fore likely to be adaptive, *i.e.*, provide a distinct functional advantage [6]. In *E. coli* duplications primarily yield obligatory homomers, with each paralog mediating a different enzymatic func-tion (typically different substrate specificity). In yeast, however, the obligatory heteromers seem to be associated with acquisition of new regulatory modes. Thus, function may dictate the fate of the oligomeric state. Another factor might be the location of the active-site that in some enzymes resides within the subunits and in others at the interface between subunits [31]. Also of note is that, in principle, divergence of a heteromeric interaction increases the likeli-hood that both copies would fix in the genome, because loss of one copy leads to non-functio-nalization. Duplication itself is random, yet whether a duplicate is fixed or lost (the far more likely fate) depends on how rapidly it provides a selectable advantage [32]. Gene knockout experiments support this hypothesis–deletion of one copy is highly deleterious in heteromers while for obligatory homomers deletion of one copy often has little effect (**S3 Data**).

Future work might address the above and other questions, and may also track down other possible evolutionary transitions–*e.g.* the dominating trend is homo-to-hetero transitions, yet can we track down cases of heteromers that diverged to homomers? Addressing these ques-tions will demand detailed phylogenies and experimental evaluation of the oligomeric states before and after the duplication. However, a rigorous way of assigning oligomeric states from molecular interaction databases, and of determining the fate of duplicates, is crucial to any such investigation.

## Methods

Further details are provided in **S1–S6 Data**, in relation to the each of analyses described therein.

### Detecting *S. cerevisiae* and *E. coli* paralogous protein pairs

The 1st step of our analysis identified all *S. cerevisiae* and *E. coli* paralogous protein pairs (**Fig 1B**). To this end, all-versus-all intra-species protein-protein BLAST [33] was performed across their respective proteomes, obtained from NCBI Genome Database [34]. BLAST hits associated with at least 25% identity and 40% query coverage were manually inspected and assigned as putative paralogous pairs (3958 pairs in *S. cerevisiae* and 2090 pairs in *E. coli*, **S1 Fig**). These pairs were further classified into three overlapping groups, with increasing stringency of paralo-gue assignment, Low-Confidence (LC, ≥25% identity, ≥40% coverage), Medium-Confidence (MC, ≥30% identity, ≥50% coverage) and High-Confidence (HC, ≥40% identity, ≥60%

coverage, and identical domain content). To ensure identical domain content, we compared the Pfam [35]–annotated domain contents of all HC pairs. Pfam uses Hidden Markov Models to identify domains and every annotated instance is given a probability score ($p$-value). Any domain assigned with $p < 10^{-5}$ significance was considered for further analysis. Following domain assignments, paralogous pairs were compared and those that differ in their domain content were discarded. The list of 455 *S. cerevisiae* ohnologs (paralogs emerging from the whole genome duplication; **S1 Fig**) were collected from the Yeast Gene Order Browser [36].

## Identifying molecular interactions

**Curated complexes.** Curated homo- and hetero-meric macromolecular complexes of both *S. cerevisiae* and *E. coli* were collected from Protein Data Bank [37], 3D complex database [38] and Complex Portal [26]. Complexes that include at least one protein annotated as paralog were classified into three groups, with increasing stringency of curation accuracy (**S1** and **S2 Data**). The first group, C complexes, comprises 127 *S. cerevisiae* and 18 *E. coli* complexes annotated in Complex Portal, for which only the subunit composition data are available (subunit stoichiometry is either unknown or only partially known). The second group, CS complexes, comprises 83 *S. cerevisiae* and 33 *E. coli* complexes annotated in Complex Portal, for which both subunit composition and stoichiometry data are available. The third group, PDB complexes, includes 167 *S. cerevisiae* and 117 *E. coli* complexes collected from the Protein Data Bank, for which subunit composition, stoichiometry as well as interaction patterns are known. The subunit stoichiometry of PDB complexes were further cross-validated by 3D complex database [38] annotations.

**Protein-protein interaction data.** For *S. cerevisiae*, 721701 binary PPI data were collected from seven different databases: BioGRID [39], DIP [40], HiNT [41], IntAct [42], iRefIndex [43], Mentha [44], and STRING [45] and 123644 interactions involving paralogous pairs were extracted. For *E. coli* 47727 binary PPI data was compiled, by adding one additional dataset [46] to the above seven databases; 4376 interactions involve paralogous proteins. Note that these PPI databases include both high- and low-throughput data, with the former dominating (see also next section). Predicted interactions, and text-mining based interactions, reported in STRING were removed. *S. cerevisiae* raw PPI data were filtered in three successive steps (**S1 Data**). In the 1st step, 90 transposon element proteins encoded by genes of viral origin annotated in the Saccharomyces Genome Database [47] were removed. In the 2nd step, to minimize false-positives in the PPI data, we demanded that the interaction between two proteins observed using both proteins as bait and as prey, and the interaction must be reported in at least two databases. The bait-and-prey information is relevant to high-throughput two-hybrid and pull down experiments, and hence this filtering criterion removed interactions detected by other means, foremost by low-throughput methods such as gel shifts. However, this filtering resulted in a negligible loss of interacting pairs and did not bias the results (see next section). Also note that the databases used here collect their raw data from published literature. Overlaps between databases are therefore common, although none of these databases overlap completely. Thus, the demand that the interaction must be reported in at least two databases does not necessarily mean two independent observations, but as a minimum it eliminates annotation mistakes. In the 3rd filtering step, interactions between two proteins localized in different sub-cellular compartments were excluded. For this, yeast protein localization data obtained from LoQAtE [48], Yeast GFP Fusion Localization Database [49] and Yeast Protein Localization database [50] were combined together. These filtering steps resulted in the final PPI dataset of 28381 pairwise interactions involving paralogous proteins (**S1 Data**). The PPI datasets derived after each step of filtering are provided in **S1 Data**. Transposon elements were

absent in the *E. coli* raw PPI data and filtering involved only one step (interactions must be reported for both proteins as bait and as prey, and in at least two databases). This yielded a final PPI dataset of 1996 pairwise interactions (**S2 Data**).

## Assigning the interaction status of paralogous pairs

Based on the interactions in the above-described molecular interaction datasets, paralogous pairs were assigned to one of the four categories described in **Fig 1A**: obligatory hetero (the two paralogs do not self-interact, but cross-react to form a heteromer), mixed homo/hetero (two paralogs cross-react to form a heteromer, and at least one paralog also self-interacts), or hetero others (only one paralog self-interacts and the other interacts with to another, non-paralogous partner). Obligatory homomers were assigned using a *stringent* and a *flexible criterion*. The *stringent criterion* demanded that the two paralogs do not cross-react, and that *both* self-interact; the *flexible criterion* demanded that the two paralogs do not cross-react and at least one of them self-interacts.

PDB structures and PPI data, by definition, comprise physical interaction data between proteins. For CS and C complexes, inter-subunit interactions were predicted from the PPI data. A homomer was assigned if it is present in multiple copies in a complex, and also self-interacts in the PPI data. Heteromers were assigned if both paralogs co-occur in a complex and found to cross-interact, but not self-interact, in the filtered PPI dataset. For obligatory homomers in the curated complexes, we also ensured that the two paralogs do not cross-interact in the PPI data.

To examine if the assignment of homo/hetero fates in the PPI dataset was substantially influenced by the bait-prey filtering, we extracted all the PPI data that were detected exclusively by methods other than two-hybrid and pull downs. As with other PPI data, these data were filtered by demanding that the interaction is reported in at least two databases, and by excluding interactions between proteins localized in different sub-cellular compartments. For the filtered subset of data, applying the *stringent criterion*, we assigned the interaction status of paralogous pairs. Only 14 new pairs were detected (**S6 Data**), compared to 1152 pairs in the filtered PPI data (see **S3 Data**), indicating that a negligible amount of PPI data was lost due to the bait-prey filtering. Further, in these 14 pairs, the overall dominance of heteromers in yeast was reflected (5 obligatory hetero, 7 mixed, 2 hetero-others, and no obligatory homo; **S6 Data**).

## *S. cerevisiae* and *E. coli* orthologous proteins

To identify the orthologous *S. cerevisiae* and *E. coli* protein pairs, inter-species reciprocal protein-protein BLAST [33] searches were performed. In total, 7325 protein pairs associated with e-value $< 10^{-5}$ were extracted. We then identified the subset of these pairs that comprise a homomeric protein in *E. coli* and an obligatory-hetero, or mixed homo-hetero paralogous protein in *S. cerevisiae*. The domain content of these pairs, as annotated in Pfam [35], were compared and those sharing at least one domain, were extracted. These pairs were then manually checked for having the same function in the two organisms and that the shared domain corresponds to this function. When consolidated, this analysis extracted orthologous relationships between 103 *E. coli* homomeric proteins (52 singletons and 144 paralogous pairs) and 421 paralogous *S. cerevisiae* proteins (235 pairs; **S5 Data**).

## Statistical analysis

All the computation and statistical analyses were performed using in-house Python codes. Graph plots were generated using OriginLab software and Adobe Photoshop.

## Supporting information

**S1 Fig. Histogram plots of *E. coli* and *S. cerevisiae* paralogous pairs binned by sequence identity. (A)** *E. coli* paralogs (n = 2090 pairs). **(B)** *S. cerevisiae* all paralogs (n = 3958 pairs). **(C)** *S. cerevisiae* ohnologs (the subset of paralogs that arose from the whole genome duplication; n = 455 pairs). Note that these plots include all paralogs, not only the ones for which molecular interaction data are available. The dotted red lines represent the identity thresholds used for defining MC ($\geq$30% identity) and HC ($\geq$40% identity).
(TIF)

**S1 Data. The *S. cerevisiae* molecular interaction dataset used in this study (including the list of the curated complexes and the PPI data).**
(XLSX)

**S2 Data. The *E. coli* molecular interaction dataset used in this study (including the list of the curated complexes and the PPI data).**
(XLSX)

**S3 Data. The inferred interaction status of *S. cerevisiae* paralogous pairs, in curated complexes and in PPI data.** For paralogous pairs in curated complexes, deletion phenotypes are also provided.
(XLSX)

**S4 Data. The inferred interaction status of *E. coli* paralogous pairs, in curated complexes and in PPI data.**
(XLSX)

**S5 Data. List of S. cerevisiae heteromeric paralogs that relate to homomeric E. coli proteins.**
(XLSX)

**S6 Data. The inferred interaction status of *S. cerevisiae* paralogous pairs in PPI data detected exclusively by methods other than two-hybrid and pulldowns.**
(XLSX)

## Acknowledgments

We sincerely thank Emmanuel Levy and Christian Landry for critically reading of an early version of this manuscript and for their valuable comments and suggestions.

## Author Contributions

**Conceptualization:** Saurav Mallik, Dan S. Tawfik.

**Data curation:** Saurav Mallik.

**Formal analysis:** Saurav Mallik.

**Funding acquisition:** Dan S. Tawfik.

**Investigation:** Saurav Mallik, Dan S. Tawfik.

**Methodology:** Saurav Mallik.

**Project administration:** Dan S. Tawfik.

**Resources:** Saurav Mallik.

**Software:** Saurav Mallik.

**Supervision:** Dan S. Tawfik.

**Validation:** Saurav Mallik, Dan S. Tawfik.

**Visualization:** Saurav Mallik.

**Writing – original draft:** Saurav Mallik.

**Writing – review & editing:** Saurav Mallik, Dan S. Tawfik.

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
