## [Decision Letter · Decision Letter 0]

15 Jun 2020

Dear Mr Tawfik,

Thank you very much for submitting your manuscript "Determining the interaction status and evolutionary fate of duplicated homomeric proteins" for consideration at PLOS Computational Biology. As with all papers reviewed by the journal, your manuscript was reviewed by members of the editorial board and by several independent reviewers. The reviewers appreciated the attention to an important topic. Based on the reviews, we are likely to accept this manuscript for publication, providing that you modify the manuscript according to the review recommendations.

This paper provides valuable insights and we will be interested to consider it further after reviewers suggestions are addressed in the revised version. One more comment: the homo- vs hetero- oligomerization paths for dimer formation were carefully considered in an earlier work Lukatsky et al JMB (2007) v.365 pp.1596-1606. This work should be cited and discussed in the present manuscript.

Sincerely,

Eugene I. Shakhnovich

Guest Editor

PLOS Computational Biology

Arne Elofsson

Deputy Editor

PLOS Computational Biology

[LINK]

This paper provides valuable insights and we will be interested to consider it further after reviewers suggestions are addressed in the revised version. One more comment: the homo- vs hetero- oligomerization paths for dimer formation were carefully considered in an earlier work Lukatsky et al JMB (2007) v.365 pp.1596-1606. This work should be cited and discussed in the present manuscript.

Reviewer's Responses to Questions

**Comments to the Authors:**

Reviewer #1: Mallik and Tawfik present a valuable computational study on the fate of duplicated oligomers. Duplications of oligomers initially lead to the formation of hetero-oligomers, and this initial situation can subsequently resolve into a variety of different scenarios, including obligate assembly into heteromers, assembly into only homomers, and losses of interfaces. Previous studies have variously estimated the frequency of these different fates, using high throughput interactomics data. This is associated with a number of biases and difficulties that Mallik and Tawfik address in this paper. One difficulty that remains (which all prior work also suffers from), is that using databases from a small number of model organisms makes it impossible to make explicit, polarized evolutionary inferences about changes in biochemical phenotypes. Nevertheless, this manuscript substantially improves on prior attempts to answer this question using similar methods. I consider this a paper valuable contribution to this question that moves the field forward.

The authors methodology is solid and the conclusions well supported as far as I can tell. My only minor comment is that perhaps Finnigan et al 2012 could be cited when the authors mention in the discussion that subfunctionalization of interfaces can aide the preservation of genes after duplication, as this idea has been articulated (and demonstrated experimentally) before.

Reviewer #2: An interesting study on the evolution of protein complex formation which will add to our understanding of the importance of gene duplication to our current understanding of systems biology. I would be interested in the answers to the following questions:

1.The statement about the reliability of Y2H as as a methodology is now 18 years out of date,according to the reference used and would certainly be challenged by groups such as Vidal or Uetz. The authors either need to find more up-to-date evidence to support that statement or qualify it to make a more current assessment of this technology.

2. What is the authors definition of 'high-throughput' PPI data? Whilst there is certainly plenty of these in both IntAct and BioGriD, both of these resources also contain significant amounts of low-throughput data, by most definitions - was this not used and, if not, then why not. The authors otherwise are largely limiting themselves to Y2H data for the identification of homomeric interactions, which seems strange if they really believe it is of poor quality.

3. For all the resources used, a list of release numbers/download dates would have been useful when I tried to understand how the data had been used.

4. The dataset filtering uses the bait-prey relationship. This is fine if the authors have limited themselves to y2H and pulldown expts, but if they used other valuable data sources such as gel filtration and comigration in gels, how did they deal with data types which do not have this relationship.

5. The authors had to skip the lack of colocalisation filter step for Ecoli - did they look at using Gene Ontology data for this. Or the colocalisation data in the IMEx databases. There may not be enough information in either case, but it would be interesting to know if the authors had considered this.

5. The authors describe 7 different databases as 'curated' - this is fact not true. iREFIndex and STRING compile manually curated data from BioGRID and the IMEx data held in IntAct but do not add any manually curated data themselves whilst MENTHA is a browser which visualizes IMEx and BioGRID data. HiNT do at least add PDB data but I am not clear what was additionally gained from iREFIndex, STRING or Mentha. If STRING predictions/text-mining data was used, this should be made clear by the authors.

6. Looking in the PSICQUIC viewer, the Rajagopala dataset is in IntAct and therefore also in Mentha and irefIndex and presumably also STRING.Why do the authors treat that set separately -it would surely be better to run it through whatever data filters were used for the interaction database sets.

7. There does not seem to have a list of complexes used for Ecoli, only the proteins.

8. I was wondering why the authors converted all the Yeast information to yeast gene identifiers rather than leave them as UniProt protein IDs, which would make them more easily comparable with the E.coli data for anyone trying to work with the data. As UniProt seems to be the identifier in common used by most of the the resources listed, including Pfam, it seems an unnecessary extra step and could lead to data loss.

**Have all data underlying the figures and results presented in the manuscript been provided?**

Reviewer #1: Yes

Reviewer #2: Yes

PLOS authors have the option to publish the peer review history of their article (what does this mean?). If published, this will include your full peer review and any attached files.

Reviewer #1: No

Reviewer #2: No
---

## [Editor Report · Decision Letter 1]

12 Jul 2020

Dear Prof. Tawfik,

We are pleased to inform you that your manuscript 'Determining the interaction status and evolutionary fate of duplicated homomeric proteins' has been provisionally accepted for publication in PLOS Computational Biology.

Best regards,

Eugene I. Shakhnovich

Guest Editor

PLOS Computational Biology

Arne Elofsson

Deputy Editor

PLOS Computational Biology

---

## [Editor Report · Acceptance letter]

18 Aug 2020

PCOMPBIOL-D-20-00771R1 

Determining the interaction status and evolutionary fate of duplicated homomeric proteins

Dear Dr Tawfik,

I am pleased to inform you that your manuscript has been formally accepted for publication in PLOS Computational Biology. Your manuscript is now with our production department and you will be notified of the publication date in due course.

With kind regards,

Matt Lyles
